# Teachers' Professional Action Competence in Education for Sustainable Development: A Systematic Review from the Perspective of Physical Education

**Julia Lohmann** [1,*] , **Jennifer Breithecker** [1] , **Ulrike Ohl** [2] , **Petra Gieß-Stüber** [3] **and Hans Peter Brandl-Bredenbeck** [1]

[1] Institute of Sport Science, University of Augsburg, Universitätsstr. 3, 86159 Augsburg, Germany; jennifer.breithecker@uni-a.de (J.B.); brandl-bredenbeck@sport.uni-augsburg.de (H.P.B.-B.)

[2] Institute of Geography, University of Augsburg, Alter Postweg 118, 86159 Augsburg, Germany; ulrike.ohl@geo.uni-augsburg.de

[3] Institute of Sport and Sport Science, Albert-Ludwigs-University of Freiburg, Sandfangweg 4, 79102 Freiburg, Germany; petra.giess-stueber@sport.uni-freiburg.de

* Correspondence: julia.lohmann@uni-a.de

**Abstract:** In order to implement education for sustainable development (ESD), teachers from all subjects need to be equipped with ESD-specific professional action competence, including physical education (PE) teachers. However, the current state of research on approaches to defining ESD-specific teacher competence is complex and there is little debate on what competences PE teachers in particular need in order to implement ESD. The purpose of this study is to make a theoretical contribution to clarifying the central concepts of ESD-specific teacher competences and to link this discussion to the subject of PE. We conducted a systematic literature review following PRISMA guidelines with a focus on normative and theoretical work about ESD-specific teacher competences. Twenty-two articles from 2008 onwards met the inclusion criteria. Subsequently, we applied a qualitative content analysis based on theoretically derived main categories. The literature review revealed a more nuanced examination of the categories of ESD-specific professional knowledge and beliefs. The categories of motivational orientation and self-regulation were found to have received less attention in the analyzed papers. PE-specific aspects were not reflected in the reviewed literature. A refined model of ESD-specific professional action competence is suggested and it is demonstrated how this model might be applied to subject-specific discourses from the perspective of PE.

**Keywords:** teacher; knowledge; belief; competence; sustainability; education; sport; review

## 1. Introduction

If education for sustainable development (ESD) is to become viable, competent teachers are needed [1]. ESD empowers learners to acquire those competences needed to understand ecological, economic, socio-cultural and political interdependencies on a local and global level, to envision possible, probable and desirable futures and to responsibly and collaboratively act towards a sustainable future [1,2]. School teachers are faced with the challenge of adding and implementing these key sustainability competences (KSCs) as cross-curricular learning objectives to their otherwise subject-specific teaching agendas. To meet this challenge, teachers need ESD-specific professional competence in addition to their subject-specific competence [3]. For example, physical education (PE) teachers not only need to be able to help their learners acquire motor skills, health-related knowledge and understanding and the rules and tactics of sport activities, but also to empower them to understand and reflect the way sport and exercise are embedded within ecological, economic and political systems. Moreover, they need to be able to develop, justify and present their own objectives for the sustainable development of sport and exercise [4].

For teacher education, the question arises as to what competences are particularly relevant for teachers in the context of ESD [5]. Numerous models for ESD competences have been developed for graduates from higher education institutions (e.g., [6]), educators in general (e.g., [7]), educators in non-formal settings [8], and for schoolteachers (e.g., [9–13]. Regarding the latter, existing models draw on different theoretical backgrounds and heuristic frameworks. They overlap to a large extent but have not yet been systematically analyzed and compiled as part of a theory-based approach. Furthermore, in the specific context of PE there is hardly any scientific debate about ESD and, in particular, about the professional competence a PE teacher needs to implement ESD in PE.

The purpose of the present research is to give an overview of existing approaches to define ESD-specific professional teacher competence and to summarize and integrate the existing approaches against the background of the generic model of professional action competence [14]. With this study we contribute to the theoretical and normative discussion about ESD-specific professional teacher competence and propose a theoretical framework that is beneficial to subject-specific discourses. Using the subject of PE as an example, we will show how ESD-specific and subject-specific professional competence could be integrated.

## 1.1. Context of This Research

This research is intended to be an international contribution to the research in teacher competences for ESD. However, the research team is located in Germany and some implicit assumptions regarding ESD and PE are potentially biased by a national perspective. Therefore, we want to roughly outline the educational policy framework for ESD in this national context. In Germany, ESD is implemented in each of the 16 federal curricula of general education schools and is anchored in legal frameworks [15]. At the curricular level, ESD is usually formulated as a cross-curricular educational objective [15]. The "Curriculum framework: Education for sustainable development" [4] plays an important role in the formulation of subject-specific learning goals and their implementation. It offers educational targets (key competences) and a didactical framework, as well as specific topics for different subjects, including PE [4]. As such, it has become one of the standard works for the implementation of ESD in Germany, for example, in teacher education or non-formal education.

Germany, in turn, is embedded in the European context. At the European level, the United Nations Economic Council for Europe (UNECE) has developed frameworks and guidance for the implementation of ESD across education sectors. The document "Learning for the future. Competences in education for sustainable development" [12] identifies core competences in ESD for educators. In line with what is outlined there, national ESD strategies with a focus on higher education have been developed in several European countries, such as Cyprus, Greece, Spain, Serbia, Slovenia, Slovak Republic, Ireland, the UK and Flanders [16]. At a global level, the lead UN agency on ESD, UNESCO, has published the "Framework for the implementation of education for sustainable development (ESD) beyond 2019" [17]. In this publication a new framework for ESD is proposed and pedagogical implications for transformative learning are addressed.

## 1.2. Defining Sustainable Development (SD)

For our paper we draw on the final report of the World Commission on Environment and Development ([18]; "Our common Future"), which is generally remembered today for a single phrase defining SD as meeting "the needs of the present without compromising the ability of future generations to meet their own needs" (para. I/3/27). Beyond this definition, the report made clear that SD may only be reached by reconciling tensions between present and future generations, economic versus environmental perspectives, North versus South, and scientific accuracy versus political acceptability [19]. Thus, SD must be shaped by individual and political decisions that consider the multidimensionality of issues, i.e., environmental, economic and socio-cultural dimensions and their interconnectedness. The

goals are to reach intergenerational justice through short- and long-term estimation of action consequences (temporal scale) and intragenerational justice through the estimation of action consequences on individual, local, regional and global levels (spatial sphere) [19,20].

*1.3. Education for Sustainable Development (ESD)*

Education plays an important role in the societal transformation that is needed for SD [21]. In this article we adopt an emancipatory perspective on ESD [22,23]. Emancipatory ESD is focused on capacity building and critical thinking rather than on instrumental goals like directly changing learners' behaviors [22]. It aims at fostering KSCs in learners, namely, systems thinking competence, anticipatory competence, normative competence, strategic competence, collaboration competence, critical thinking competence, self-awareness competence and the overarching integrated problem-solving competence [2].

Defining the term competence for this study, we refer to the functional-pragmatic concept of competence [24]. Klieme et al. [24] define competences as "context-specific cognitive dispositions that are acquired by learning and needed to successfully cope with certain situations or tasks in specific domains" (p. 9). Competences may refer to very narrowly defined domains or situations or be defined more broadly in terms of overarching key competences [24,25]. In the literature, the term 'competency' is often used interchangeably with 'competence' and they are often used inconsistently or synonymously [26].

From a didactical perspective, ESD should be based on a holistic (environmental, social, economic and political, on local and global scales), pluralistic (considering different perspectives and values, fostering critical thinking) and action-oriented teaching approach [27]. Thus, ESD goes beyond environmental education which is usually focused on ecological aspects and pro-environmental behavior or global citizenship education which centers around the concept of citizenship (social and political dimensions). For teachers, the competence to deal with complexity is one of the most significant challenges of teaching ESD [28]. Learning situations in ESD are often characterized by a double complexity [29]: *factual* complexity refers to the complexity of the facts; *ethical* complexity refers to moral uncertainties and conflicts of goals. Ohl [28] identified three relevant fields of action for educators in order to face the didactical challenge of complexity.

- Acquisition of well-founded (scientific) knowledge: Educators should reflect the complexity and controversies in given issues, use strategies to reduce and order complexity (i.e., key concepts), critically evaluate information and show how to cope with uncertain knowledge.
- Highlighting the relevance and implications of an issue for different stakeholders and working out political, social and individual options for action: Educators should explore options for action together with learners from multiple and controversial perspectives without determining concrete guidelines for action or specific behavioral requirements. Teachers must therefore be sensitive when expressing their personal views and convictions and know the limits and power of their influence.
- Using scientific knowledge and value scales for the assessment of a situation and the options for action: Educators should support learners in producing knowledge- and value-based arguments to justify their individual decisions and actions.

Teachers play a prominent role in the education process regarding learning outcomes [30] and initiating change in schools [31]. Through a cascade of chain links, the professional competence of a teacher may be linked to student characteristics and learning outcomes [32]. For ESD, UNECE [12] recognized early on that the ESD-specific competence of educators is a "bottleneck in achieving ESD" (p. 7), and the competence-oriented education of teachers and other multipliers should be given high priority.

*1.4. Existing Approaches for Teachers' ESD-Specific Professional Competence*

Several approaches exist to describe competences teachers need in order to implement ESD. However, the current state of research on ESD-specific teacher competence is complex and is in part "dominated by 'laundry lists' of competencies (sic.)" [6] (p. 204)

and beset by a "lack of justification linking theoretical frameworks with their practical implementation" [13] (p. 6). The existing literature ranges from short lists of overarching KSCs (e.g., [33]) to extensive (and sometimes redundant) lists or frameworks of complex competences (e.g., [11,34]). KSCs usually refer to learning outcomes at the pupils' level. However, some researchers also define KSCs as learning outcomes for teacher education programmes [33].

It is only in more recent studies that the educational science debate on the professionalization of teachers [14,32,35] is taken up in the context of ESD. Starting from the subject of mathematics, [14] developed a generic model of professional action competence (PAC) that has been theoretically reflected and applied in empirical studies in several domains, including PE [36,37], maths, science and language [32] and also in the wider context of ESD [5,10,38,39]. According to Baumert and Kunter [14], professional teaching practice is a result of the interplay between professional knowledge, beliefs, motivation, and self-regulation. *Professional knowledge* stands at the core of PAC. Most important for teaching practice are domain-specific content knowledge (CK) and pedagogical content knowledge (PCK). Additionally, teachers need domain-general knowledge about how to shape the process of teaching, e.g., general pedagogical knowledge, institutional knowledge or counselling knowledge. *Beliefs* and knowledge differ in terms of their epistemological status [14]. They should therefore be conceptualized as separate categories of teacher competence [14]. Values and beliefs that are important for teaching practice include "value commitments, epistemological beliefs (world views), subjective theories of teaching and learning, and goal systems" [14] (p. 37). *Motivation* and *self-regulation* abilities are important determinants of intentions and behavior and are therefore relevant for psychological functioning [14]. This model of teachers' PAC reflects a rather narrow understanding of competence, i.e., it is conceptualized as profession-specific (i.e., specific for schoolteachers) and domain-specific (i.e., PE, science or ESD).

The various approaches to ESD-specific teacher competence are difficult to compare and overlap to a large extent. However, they have not yet been systematically analyzed and summarized. Therefore, the first purpose of this study is to give an overview of existing approaches to ESD-specific teacher competence, to summarize their subcomponents and to integrate them into one model that can be linked to domain-specific discourses, such as PE [3], or other subjects and subject areas [40].

### 1.5. Physical Education Teacher Education in the Context of ESD

Physical education (PE) holds great potential to contribute to the learning goals for ESD. This potential has so far been neglected in the scientific literature [3]. PE is body-centered and therefore has a unique position in the school curriculum [4]. Additionally, many young people express high motivation for this subject [41]. Thus, sports and games may be the starting point for differentiated experiences with and reflections about individual sustainable lifestyle choices relating to physical activity [42,43] and about social, economic, ecological and political entanglements in local and global sport sectors [4]. In PE, holistic work with body and mind can increase mindfulness [44], which in turn is linked to sustainable behaviors [45]. PE may also contribute to intercultural learning [46,47] and thus might be a path to fostering ESD learning goals, such as collaboration competence.

Teachers from all subjects, including PE, must be prepared to teach their respective subjects according to ESD goals [1,15,40]. Therefore, PE teachers have the mission to foster KSC in their students through sport, games and movement. However, prospective PE teachers are often faced with the challenge of integrating their (felt) role as athlete or coach and the PE teacher role [48–51]. With regard to ESD, PE teachers have to integrate another role into their professional self: the role of an ESD educator [3]. However, there is little scientific debate on what PE teachers need to know about issues of sustainable development, the teaching of these issues in the PE context, and which beliefs, motivational orientations and self-regulations are helpful to implement emancipatory ESD in PE.

Therefore, the second purpose of this paper is to discuss ESD-specific professional teacher competence from the perspective of PE and to give examples of an integrated ESD- and PE-specific professional teacher competence.

### 1.6. Purpose of This Study

The purpose of this study is to make a contribution to clarifying central concepts and to furthering the discussion of the professionalization of schoolteachers in the field of ESD using the example of PE. To reach this goal we first analyze and evaluate the core attributes of the professional competence construct for teachers in the context of ESD, as reflected in contemporary research literature (up to August 2020). We synthesize findings and summarize what the existing literature tells us about the professional knowledge, beliefs, motivation and self-regulation teachers should acquire to implement ESD. Subsequently, we discuss the findings from a PE perspective.

This paper will explore the following research questions:

(1) What approaches are used to describe ESD-specific teacher competence?
(2) What are the defining attributes of ESD-specific professional teacher competence in the context of ESD?
(3) How may ESD-specific competences be integrated into the professional self of PE teachers?

Research questions (1) and (2) will be addressed through a systematic literature review, as detailed in the Methods and Results sections. Question (3) will then be addressed in the discussion.

## 2. Methods

An electronic database search strategy was employed using the following databases: (i) Web of Science core collection, (ii) ERIC and (iii) ScienceDirect for English literature; and (iv) peDOCS, (v) fachportal-paedagogik and (vi) BISp-SURF for German literature. We adopted the year 2000 as the start date and the last search was conducted on the 13 August 2020. These education, sports and PE and social science databases are suitable for the topic at hand and increase the probability that relevant papers have been located [52]. The three key concepts 'professional competence', 'education for sustainable development' and 'teacher' were included in the search and combined with the Boolean operator AND [53]. For each key concept, synonyms and related terms were specified and combined with OR. The key concept 'physical education' was included in the search term at the beginning as well. However, searches including this key concept did not reveal any records. Therefore, we continued the search without the term 'physical education' and its synonyms. 'English', 'German', 'book', 'journal article', and 'dissertation' filter boxes were marked on all searches if applicable. For more details about the search strategy and search history, see Supplement A.

### 2.1. Eligibility Criteria

The criteria for inclusion in this systematic review were:

(i) Original papers that clearly define or describe one or more ESD-specific teacher competence(s) (e.g., ESD-specific professional action competence, KSC on the teacher level)
(ii) Papers that refer to ESD as a basic concept, i.e., multidimensional (ecological, social, economic), taking into account local and global development, today and in the future

To address the aims and research questions, the following exclusion criteria were adopted for the database search:

(i) Papers not covering the definition of single components of the professional competence of teachers in the context of ESD;
(ii) Papers that use existing definitions of ESD-specific teacher competence to support empirical research without furthering, expanding or integrating new ideas to the existing model.

(iii) Papers that only refer to the ecological dimension or strongly focus on the social and cultural dimension while overlooking the multidimensional character of the ESD concept;

(iv) Papers referring to KSC as learning outcomes at the pupils' level, learning outcomes of higher education in general or competences of academics in higher education or the informal education setting without specifying the profession of teachers;

(v) Book reviews and book synopses; conference reports and readings; editorials and forewords; grey literature and brochures.

The authors followed the preferred reporting items for systematic reviews and meta-analyses (PRISMA-P) guidelines [54]. The database search revealed 1025 records. In accordance with the PRISMA procedures, 332 duplicate papers were removed. Non-duplicated papers were then processed in several steps. First, titles and abstracts were screened pertaining to the inclusion and exclusion criteria by two independent reviewers. Second, full texts were obtained for records that met the basic inclusion criteria and these were read thoroughly and deemed either suitable or unsuitable according to the inclusion and exclusion criteria by two independent reviewers. In both steps, any discrepancies between reviewers were resolved by consensus. Records were kept of this process, with 88% agreement prior to discussion and 100% post discussion for the screening of titles and abstracts and with 73% agreement prior to discussion and 100% post discussion for the full-text analysis. After conducting the database search, a secondary manual search was conducted. The Journal of Teacher Education for Sustainability was searched in the years 2018–2020 to doublecheck the database search. This search did not reveal any new publications that met the eligibility criteria. Additionally, a manual search in the reference lists of publications obtained through the database search was conducted and revealed another six relevant papers. During the data analyzing process, the research team performed the following roles: main analyst (J.L.), coanalysts (J.B., H.K., M.W.) and critical colleagues (H.P.B.-B., U.O., P.G.-S.). After this process, a total of 22 papers from 2008 onwards were included in the review (see Figure 1).

### 2.2. Appraising the Quality of Studies

The quality of the publications was categorized into three levels based on an adapted version of the JBI critical appraisal checklist for text and opinion papers [55] (see Table 1 for an overview; see Supplement B for the adapted critical appraisal checklist). The quality appraisal was conducted by two reviewers (J.L., J.B.) independently; inconsistencies were discussed and resolved by consensus.

- Low quality: Competences are listed as a 'laundry list' without logical structure and a traceable analytical process. There is no critical examination of the suggested competence model and it is not systematically embedded in relevant literature.
- Medium quality: Competences are described normatively based on the literature and structured within a heuristic or logical model. In most cases, reference is made to relevant literature. However, the literature has not been systematically evaluated and often there is no critical appraisal of the suggested heuristic or model.
- High quality: Some research groups made the effort of structured expert consultation and systematically summarized ESD-specific teacher competences. High quality papers report a logical structure of competences and an analytical process for defining these competences. The suggested competence models are systematically embedded in the literature. In some cases, authors critically examine their own approach.

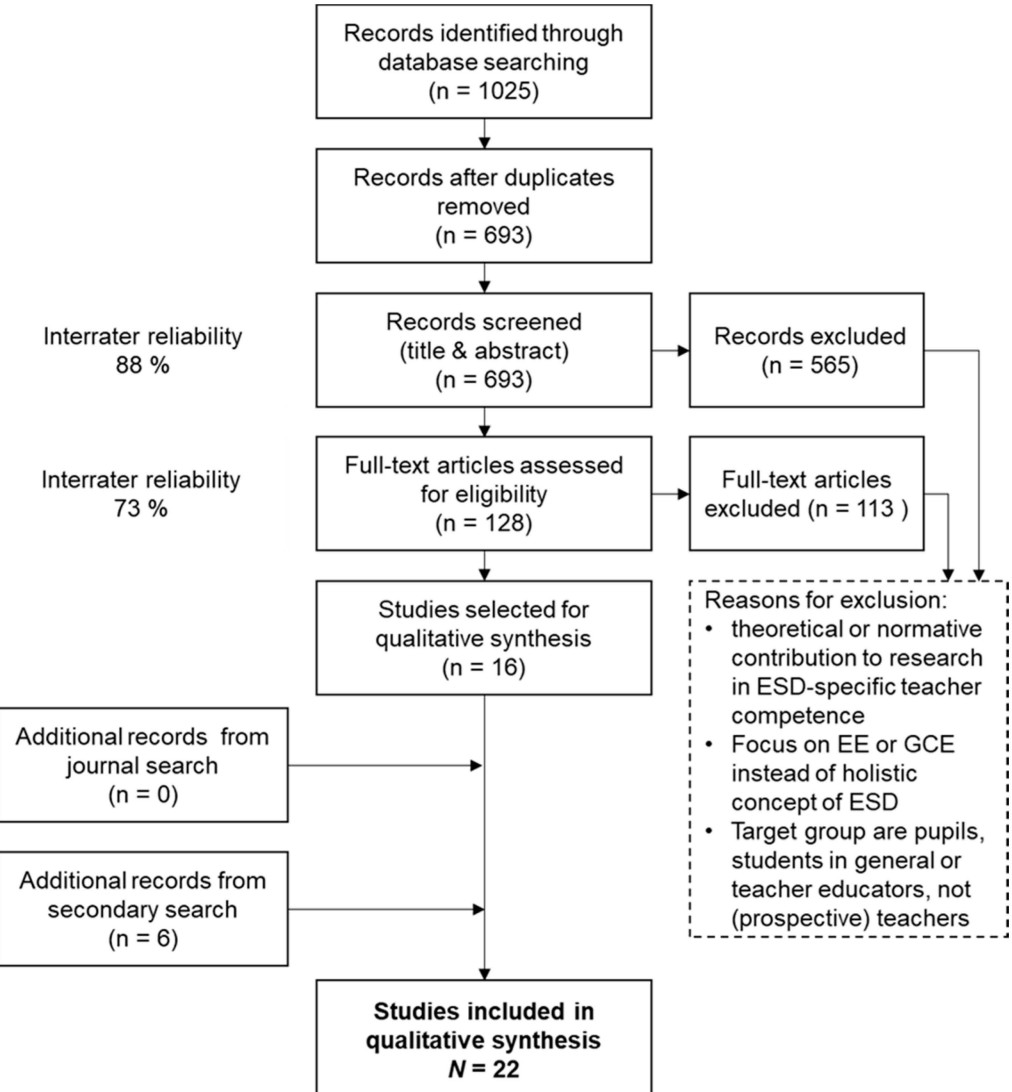

**Figure 1.** Flow diagram.

Two publications by Sleurs [11] and UNECE [12] marked the beginning of the scientific debate about teacher competence in ESD. These approaches to defining teacher competence have been used as a basis in many subsequent studies. More elaborate frameworks (e.g., [9,13]) and research based on the PAC model (e.g., [10]) will likely play a greater role in the future.

### 2.3. Data Analysis

The selected 22 papers were analyzed using qualitative content analysis [56]. This method involved a deductive definition of the main categories based on the generic model of professional action competence (PAC) [14], i.e., content knowledge (CK), pedagogical content knowledge (PCK), institutional context knowledge (ICK), beliefs, motivational orientation and self-regulation. We additionally deductively defined one main category for the approaches to defining competence and one main category with respective subcategories for key sustainability competences (KSC) [2] at the teacher level. KSCs are important for every person to act in favor of SD, thus they are important overarching competences for teachers as well. They are mentioned as important precursors of teaching ESD in some publications but they are not necessarily specific to the teaching profession. Subsequently, the deductive categories were further refined and, if appropriate, subcategories were created inductively based on the material.

Data analysis comprised the following steps: (1) coding of the entire data using the main categories, (2) compilation of all coded text passages with the same main categories, (3) inductive definition of additional categories based on the materials, (4) coding of the entire data using the refined category system (see Figure 2) and (5) evaluation and interpretation [56].

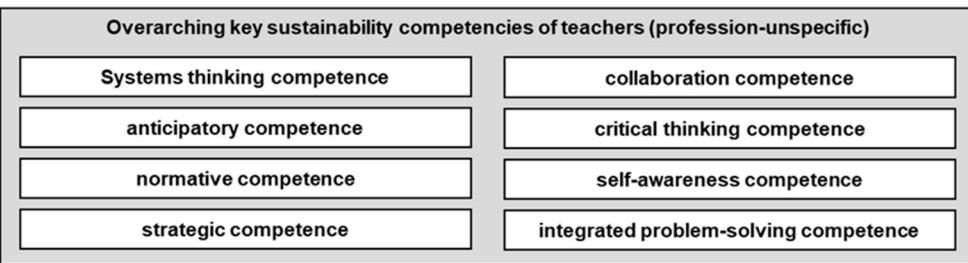

| **Definition of basic terms** | |
|---|---|
| **Approaches to define competences** | KSC: Key sustainability competences |
| | UNECE: Model by expert group of UNECE |
| | CSCT: Curriculum, SD, Competencies, Teacher training |
| | EDINSOST: Education and Social Innovation for Sustainability |
| | PAC: Generic model of professional action competence |

| **Definition of ESD-specific professional action competence (profession-specific)** | | |
|---|---|---|
| **Professional knowledge** | **Content knowledge (CK)** | Sustainability knowledge |
| | | Understanding of systems and their dynamics |
| | | Understanding of the concept of SD |
| | | Knowledge about values and emotions in context of SD |
| | **Pedagogical content knowledge (PCK)** | Knowledge of ESD-specific teaching principles |
| | | Knowledge of ESD-specific methods |
| | | Knowledge of assessment in the context of ESD |
| | | Knowledge of students's thinking related to SD |
| | | Knowledge of curriculum and resources |
| | **Institutional context knowledge (ICK)** | Fostering institutional change |
| | | Cooperation and communication |
| | | Knowledge of socio-ecological impact of education |
| | | Sustainable use of resources in educational setting |
| **Beliefs and values** | Subjective theory of and attitude towards SD | |
| | Subjective theory of and attitude towards ESD | |
| | Epistemological beliefs about knowledge | |
| | Self-perception and self-reflection | |
| **Motivational orientations** | Self-efficacy for teaching ESD | |
| | Intrinsic motivation and enthusiasm | |
| **Self-regulation** | Managing one's own emotions | |

| **Overarching key sustainability competencies of teachers (profession-unspecific)** | |
|---|---|
| Systems thinking competence | collaboration competence |
| anticipatory competence | critical thinking competence |
| normative competence | self-awareness competence |
| strategic competence | integrated problem-solving competence |

**Figure 2.** Category system.

Interrater reliability was assessed based on two papers (10% of the material) that were independently coded by two raters using the refined category system. The Kappa coefficient [57] of 0.89 indicates good interrater reliability.

## 3. Results

### 3.1. Overview of Review Sample

The final sample consists of 22 papers published between 2008 and 2020 (see Table 1 for an overview). Most publications were published as scientific papers in journals (*n* = 18) or books (*n* = 1), two are political documents (UNECE, 2012; UNESCO, 2017) and one is a report from a European research consortium [11]. Most publications (*n* = 20) were written in the European context. Only two papers were written in the context of the United States of America. Papers from Asian–Pacific, African and South American contexts did not meet the inclusion criteria. Eleven papers did not specify the school level at which teachers work or they indicated that the model would be appropriate for all levels. Eleven papers refer to teachers of elementary education and four to teachers in secondary education. Twelve publications are normative contributions without further empirical investigation. Six publications use their competence frameworks for case studies: specific courses or programs are designed based on and evaluated by means of the respective framework. For these evaluations researchers usually use expert ratings or simple self-assessments of (prospective) teachers' competence. Four studies in this review provide quantitative empirical evidence for the structure of subcomponents of ESD-specific professional competence or associations among subcomponents.

**Table 1.** Overview of included studies (alphabetical order).

| No. | Authors | Year | Origin | Journal | Search | School Level | Quality Appraisal | Empirical Evidence | Approaches to Define Competence |
|---|---|---|---|---|---|---|---|---|---|
| [58] | Albareda-Tiana & Alférez-Villarreal | 2016 | Spain | International Journal of Sustainability in Higher Education | db | General | Medium | Case study | CSCT, UNECE |
| [59] | Albareda-Tiana et al. | 2018 | Spain | Sustainability | m | Elementary | High | Case study | EDINSOST |
| [9] | Albareda-Tiana et al. | 2020 | Spain | Book chapter | m | General | High | No | EDINSOST (orig.) |
| [60] | Barth | 2016 | Germany | Beiträge zur Lehrerinnen- und Lehrerbildung | db | Elementary | Medium | No | PAC |
| [10] | Bertschy et al. | 2013 | Switzerland | Sustainability | db | Elementary | Medium | No | PAC |
| [61] | Cebrián & Junyent | 2015 | Spain | Sustainability | db | Elementary | Low | No | KSC |
| [62] | Foley et al. | 2017 | USA | Journal of Education for Sustainable Development | db | Elementary | Medium | Case study | KSC, PAC |
| [63] | Fuertes-Camacho et al. | 2019 | Spain | Sustainability | db | Elementary | Medium | Case study | EDINSOST |
| [34] | Garcia et al. | 2017 | Spain | International Journal of Sustainability in Higher Education | db | General | Medium | No | UNECE, (CSCT) |
| [5] | Hellberg-Rode & Schrüfer | 2016 | Germany | Zeitschrift für Didaktik der Biologie | db | General | High | No | PAC |

| No. | Authors | Year | Origin | Journal | Search | School Level | Quality Appraisal | Empirical Evidence | Approaches to Define Competence |
|---|---|---|---|---|---|---|---|---|---|
| [64] | Hoppe et al. | 2020 | Germany | Sustainability | db | Elementary, lower and higher secondary | High | Cross-sectional explorative study | PAC |
| [65] | Malandrakis et al. | 2019 | Greece | Journal of Enviornmental Education | db | Elementary | High | Scale validation study | PAC |
| [33] | Murphy et al. | 2020 | Ireland | Environmental Education Research | db | Elementary, lower secondary | Medium | Case study | KSC |
| [66] | Perry | 2013 | USA | Multicultural Education | db | General | Medium | No | PAC |
| [67] | Poza-Vilches et al. | 2019 | Spain | Sustainability | db | Elementary | Low | Case study | KSC |
| [68] | Rauch & Steiner | 2013 | Austria | Center for Educational Policy Studies Journal | db | General | Medium | No | CSCT |
| [69] | Rosenkränzer et al. | 2017 | Germany | International Journal of Science Education | db | Elementary, lower secondary | High | Intervention study | PAC |
| [70] | Rosenkränzer et al. | 2016 | Germany | Higher Education Studies | db | Lower secondary | High | Intervention study | PAC |
| [11] | Sleurs | 2008 | Europe | Grey literature | m | General | Medium | No | CSCT (orig.) |
| [12] | UNECE | 2012 | Europe | Grey literature | m | General | Low | No | UNECE (orig.) |
| [1] | UNESCO | 2017 | Europe | Grey literature | m | General | Low | No | Without framework |
| [13] | Vare et al. | 2019 | Europe | Sustainability | m | General | High | No | UNECE |

Note: basic models: UNECE framework developed by an UNECE expert group [12]; CSCT, Curriculum, Sustainable development, Competences, Teacher training framework [11]; PAC, professional action competence [14]; KSC, key sustainability competences [2,6]; EDINSOST, framework of the Education and Social Innovation for Sustainability project [9]; orig. originally developed and first presented within this publication. Search: db, database, m manual. Quality appraisal based on an adapted version of the JBI critical appraisal checklist for text and opinion papers [55] (see Supplement B).

### 3.2. Approaches to Defining Competence (RQ1)

In the reviewed publications, different understandings of competence were used when developing lists or frameworks of ESD-specific professional teacher competence. Basically, five different approaches can be distinguished (see Table 1 for an overview). All approaches are domain-specific (domain: ESD) but they differ in their degree of specificity regarding the profession of teachers and concrete teaching situations.

- Key sustainability competences (KSC): This is the least specific approach to defining teacher competence in the field of ESD. It means that general KSCs are listed with reference to Rieckmann [2], Wiek [6] or similar publications. KSCs are generally relevant for thinking about and acting towards SD [2]. We classify them as overarching or basic competences everybody should acquire, including teachers, but they are more distal to the specific teaching context. Supplement C gives an overview of how often KSCs were mentioned in the reviewed publications. The KSC approach reflects a broad understanding of the competence construct and lacks specific competences teachers need to master specific teaching situations.

- UNECE: An expert group of UNECE [12] suggested a list of core competences in ESD for educators at all levels of education. The listed competences are clustered around central learning experiences (learning to know, learning to do, learning to live together, learning to be) and reflect a holistic approach to ESD. However, with the broad target group of "all educators" it adopts a rather broad understanding of competence, including overarching KSCs as well as some rather specific competences for the educational context.

- CSCT (Curriculum, Sustainable Development, Competencies, Teacher training): This framework, developed within the ENSI network (Environment and School Initiatives, www.ensi.org (accessed on: 30 October 2021)) [11] and adopted by Rauch and Steiner [68], conceptualizes the teacher as an individual who is in a dynamic relationship with her or his students, colleagues and the broader society. It is a profession-specific approach relating to schoolteachers. However, it is "not based on individuals, but on a group whose members pool their competencies (sic.) for ESD in specific projects or issues and act as a team" [68] (p. 16). Due to its complex nature, it lacks clarification regarding specific competences necessary for planning and implementing concrete learning situations by individual teachers [10].
- EDINSOST (Education and Social Innovation for Sustainability): This competence map of sustainability for education degrees was developed in three phases with stakeholders from several Spanish universities [9]. It is a combination of four generic and cross-curricular sustainability competences and three levels of competence acquisition. The EDINSOST competence map is based on a rather narrow understanding of competence, i.e., it is designed specifically for (prospective) teachers. However, subcomponents often focus on the socio-ecological effect of educational activities rather than on specific knowledge and abilities that the teacher needs to design those activities.
- PAC (generic model of professional action competence): This approach refers to a model that is very well supported theoretically and empirically in several domains [32]. Papers that use this approach reflect a rather narrow understanding of competence, i.e., a profession-specific (schoolteachers) and domain-specific (ESD) understanding. Some papers taking this approach offer normative proposals, differentiating and describing single subcomponents of ESD-specific PAC (e.g., [10,66]). Others set out to operationalize and empirically investigate specific subcomponents, e.g., diagnostic skills in the context of ESD [64] or PCK for fostering systems [69,70].

### 3.3. Synthesis of ESD-Specific Professional Action Competence (RQ2)

The results of the qualitative content analysis of ESD-specific PAC will be structured according to the main categories of professional knowledge (CK, PCK, ICK), beliefs and values, motivational orientations and self-regulation. Some publications ([9,12,34,59,69,70]) are (almost) identical in wording. In these cases, we only quote the original publication. The focus is on a qualitative analysis and compilation of the subcomponents of PAC. Therefore, we do not include quantitative data in the text. Supplement C lists the number of codings per publication for each category. Tables 2–6 give an overview of the qualitative results.

#### 3.3.1. Content Knowledge (CK)

(a)   Sustainability knowledge

This category refers to the scientific knowledge of sustainability issues (Table 2). It emphasizes the importance of factual disciplinary and interdisciplinary knowledge about ecological, economic and socio-cultural aspects of global change. Rosenkränzer et al. [70] (p. 158) term this sustainability knowledge "static knowledge" and distinguish it from declarative and procedural knowledge about systems and their dynamics, which we will reflect on in the following paragraph.

(b)   Understanding of systems and their dynamics

This category is closely linked to the previous category. However, the focus here is on knowledge about "the relationship between environmental and social systems and the ability to see connections between components and patterns across temporal and spatial domains" [13] (p. 10), about concepts and principles within systems science [69,70] and about the limits of systems science and the existence of uncertainty [13].

(c)   Understanding of the concept of sustainable development

Teachers should have profound knowledge about the concept of SD, including its definition, its dimensions (e.g., ecological, economic, social, political), structuring models

(e.g., triangle, rectangle), and principles (e.g., precautionary principle, intra- and intergenerational solidarity). Teachers should be aware that SD is an evolving concept [12] and be familiar with current "national and international policy documents relating to SD and ESD" [11] (p. 48).

(d)　　Knowledge of values and emotions in the context of sustainable development

　　　　Teachers need to know "how problems are embedded with values, needs and motives" [13] (p. 20) in order to suggest and claim arguments not only based on knowledge and facts but also on ethical considerations.

**Table 2.** Subcomponents of ESD-specific content knowledge (CK) as part of ESD-specific professional action competence.

| Content Knowledge (CK)—Subcategories | Example Quotes (Reference/Page) | References |
|---|---|---|
| **Sustainability knowledge**<br>- Knowledge of challenges and problems of SD<br>- Knowledge of the causal dimension of SD problems and solutions<br>- Disciplinary, interdisciplinary, transdisciplinary knowledge<br>- Knowledge of problem-solving approaches<br>- Knowledge of goal conflicts<br>- Knowledge of the concept of citizenship<br>- Knowledge of power relations | "take disciplinary, interdisciplinary and transdisciplinary perspectives on issues of global change and their local manifestations" ([1]/52)<br>"knowledge of socially discussed problem-solving approaches and strategies" ([5]/21)<br>"ability to recognize conflicts of goals and interests of agents in a field relevant to ESD" ([10]/4)<br>"Teachers should be able to focus on understanding the concept of European citizenship, including the rights and responsibilities it confers." ([11]/55) | [1,5,10–12,34,58,60,62, 63,65,69,70] |
| **Understanding of systems and their behavior**<br>- Knowledge about relationships, multiple influences and interactions within and between ecological, social and economic systems<br>- Declarative knowledge about concepts and principles within systems science<br>- Procedural knowledge about actions or manipulations within the systems science<br>- Knowledge about visualization and explanation of systems<br>- Knowledge about the limits of systems science and the existence of uncertainty | "Basic knowledge about systems is required if one is to understand global challenges such as climate change. Dealing with complex 'wicked problems' requires a critical understanding of the relationship between environmental and social systems and the ability to see connections between components and patterns across temporal and spatial domains . . . to recognize . . . that here may be implications for our actions which are not foreseen" ([13]/10)<br>"distinguish static knowledge about ecological facts and concepts and principles that apply within systems sciences (declarative knowledge) from procedural knowledge, which contains actions or manipulations that are valid within the systems sciences" ([70]/158–159) | [5,9,11–13,34,59,68–70] |
| **Understanding of the concept of SD**<br>- Knowledge of the concept of SD (definition, model, ethical principles)<br>- Knowledge that SD is an evolving and normative concept<br>- Knowledge of relevant policy documents | "knowledge of the concept of 'sustainable development' with its basic dimensions and principles . . . sustainability triangle/rectangle as a basic structuring principle" ([5]/21)<br>"That sustainable development is an evolving concept" (20/14)<br>"The teacher knows . . . the most relevant national and international policy documents relating to SD and ESD" ([11]/48) | [1,5,9,11,12,59,65,66, 68] |
| **Knowledge about values and emotions in the context of SD**<br>- Knowledge about ethical and value discourses<br>- Knowledge about emotions and their relevance in context of SD | Understanding of concepts like "individualism, mechanism, progress, rationalism, . . . and how they become taken-for-granted views of the world and practices by which we live" ([66]/50)<br>"The educator should be aware . . . of the impact of emotions on perceptions, judgements, decisions and actions" ([34]/781). | [5,11,13,34,66] |

3.3.2. Pedagogical Content Knowledge (PCK)

(a)     Knowledge of ESD-specific teaching principles

The inductive content analysis of teaching principles in the review sample revealed a set of nine ESD-specific teaching principles (Table 3). According to the analyzed papers, adherence to these principles should empower learners "to take independently justified decisions based on differentiated knowledge and reflected values" [68] (p. 20). A differentiated explanation of each of these principles may be found in Supplement D.

(b)     Knowledge of specific ESD methods

In order to apply the teaching principles, teachers must have a broad repertoire of methods at their disposal. Concrete methods mentioned in the reviewed publications range from visualization methods to enquiry-based and collaborative learning methods (e.g., role play, field trips, case studies projects, discussion forums). The knowledge of instructional strategies additionally includes the ability to "further develop these methods themselves" [68] (p. 19) in order to align with specific goals and contents.

(c)     Knowledge of assessment in the context of ESD

Assessment competence is needed for adaptive teaching and for the assessment of "learning results in terms of changes and achievements in relation to SD" [34] (p. 781). The assessment might happen "on the fly" [64] (p. 3) or purposefully and preparedly.

(d)     Knowledge of students' thinking relating to SD

Knowledge about students was mentioned in several papers as a relevant aspect of ESD-specific teaching competence and discussed more deeply in the paper by Hoppe et al. [64] in conjunction with assessment competence. Teachers should be able to analyse their students' preconceptions, prior knowledge, beliefs, reasoning and perspectives [64,66] and tailor their teaching to their students' experiences. Adaptive teaching is thought to be an important "basis for transformation" [12] (p. 14).

(e)     Knowledge of curriculum and resources

Knowledge of curriculum and resources includes knowledge about the concept of ESD, the ability to select ESD-specific learning goals and fit them into a subject or school curriculum and knowledge about educational resources as well as relevant policy documents.

**Table 3.** Subcomponents of ESD-specific pedagogical content knowledge (PCK) as part of ESD-specific professional action competence.

| Pedagogical Content Knowledge (PCK)—Subcategories | Example Quotes (Reference/Page) | References |
|---|---|---|
| **Knowledge of ESD-specific teaching principles**<br><br>- Create participative learning environments<br>- Create an appreciative atmosphere respecting diversity<br>- Use the knowledge base from multiple disciplines, take into account multiple perspectives<br>- Use real world problems to create learning tasks<br>- Foster critical thinking<br>- Expose learners to uncertainty, dilemmas and conflicts of interests<br>- Integrate values into teaching and make assumed norms explicit<br>- Inspire creativity and innovation | "Ability to develop and provide efficient learning opportunities concerning the qualification for participation" ([10]/4)<br>"The educator must help students to clarify their own worldview and that of others through dialogue and recognize that there are different strategies" ([34]/780)<br>"The teachers focus on the action-orientation and contextualisation of the contents." ([68]/18)<br>"The teacher is able to helping learners develop critical understandings of sustainable development." ([11]/55)<br>"This requires educators to expose their learners to ethical dilemmas and leading them to think deeply about them" ([13]/12) | [1,5,9–13,34,59–61,63–65,68] |

**Table 3.** *Cont.*

| Pedagogical Content Knowledge (PCK)—Subcategories | Example Quotes (Reference/Page) | References |
|---|---|---|
| **Knowledge of ESD-specific methods, e.g.,**<br>- Visualization, illustration, representation, analogies<br>- Simulations, role play games, field trips and case studies<br>- Projects and real-world engagements<br>- Discussion forums, exhibitions, presentations, performances | "use appropriate teaching methods for EE/ES (e.g., field trips, problem solving, etc.)" ([65]/appendix, p. 3) "repertoire of ESD-specific methods (e.g., simulation models, role play, case studies, . . . )" ([5]/23) | [5,11,13,34,65, 68–70] |
| **Knowledge of assessment in the context of ESD**<br>- General: ability to assess changes and achievements regarding learning goals (of ESD)<br>- Ability to analyze what students say to diagnose preconceptions<br>- Ability to intentionally create diagnosing opportunities<br>- Use of multiple and appropriate evaluation methods | "involve more than a measurement of knowledge of sustainability concepts gained. Rather, it would require a long-term demonstration of applied understanding of knowledge of sustainability in multiple contexts. However, shorter-term forms of assessment including projects and portfolios could be effectively used to measure student learning of sustainability concepts and enactment of related practices" ([66]/50) | [1,5,34,64– 66,68] |
| **Knowledge of students' thinking related to SD**<br>- Ability to analyse students' preconceptions, prior knowledge, reasoning and perspectives<br>- Ability to teach adaptively and take students' experiences into account | "To be able to encourage students to reconstruct their alternative conceptions about ecological concepts towards valid scientific conceptions, teachers should link their instruction directly to their students' pre-existing conceptions" ([64]/2) | [11,12,34,64– 66,68–70] |
| **Knowledge of curriculum and resources**<br>- Knowledge about the concept of ESD, including learning goals and possible teaching topics<br>- Ability to select ESD-specific learning goals that fit in subject and school curricula<br>- Knowledge about educational resources and relevant policy documents regarding ESD | "Ability to choose possible teaching topics and to evaluate their aptitude for ESD regarding their economic, ecological, social and cultural design as well as their relevance for sustainability" ([10]/4) "teachers must be familiarized with these resources and guided in reflection on how they can be incorporated into the adopted curriculum of their states and local districts." ([66]/50) | [1,9–11,58,59, 63,65,66,68–70] |

*3.4. Institutional Context Knowledge (ICK)*

(a)    Fostering institutional change

Several papers emphasize the importance of embracing a whole institution approach when implementing ESD [13]. Therefore, teachers should act as change agents at school level and work together in the development of a school curriculum that implements the learning goals of ESD at the school level (Table 4).

(b)    Cooperation and communication

This category includes three facets of cooperation and communication skills beyond general cooperation and communication skills. Teachers should be able to cooperate and communicate effectively with colleagues in order "to foster transdisciplinary learning" [60] (author translation). Furthermore, it is important to be able to establish and facilitate networks with external partners and to include them in ESD-activities in school. Finally, and linked to the ESD-specific teaching principles, researchers emphasize the importance of building positive relationships with students as a basis for participatory and action-oriented teaching and learning.

(c)    Estimating the socio-ecological impact of education

The competence map developed in the EDINSOST project [9,59]) put an emphasis on the ability of teachers to estimate the socio-ecological impact of education and mitigate negative impacts or create positive impacts on SD through education.

(d)    Sustainable use of resources in the educational context

In the context of ESD, the teacher should not only teach and talk about issues of SD but also act as a role model regarding the sustainable use of resources in the school setting [63]. Only then can ESD be communicated authentically.

**Table 4.** Subcomponents of ESD-specific institutional context knowledge (ICK) as part of ESD-specific professional action competence.

| Institutional Context Knowledge (ICK)—Subcategories | Example Quotes (Reference/Page) | References |
|---|---|---|
| **Fostering institutional change**<br>-    Development of school curriculum<br>-    Fostering institutional change through whole institution approach | "Act as a change agent in a process of organizational learning that advances their school towards sustainable development" ([1]/52)<br>"Challenge unsustainable practices across educational systems, including at the institutional level" ([12]/14) | [1,11,12] |
| **Cooperation and communication**<br>-    General cooperation and communication skills<br>-    Cooperation and communication with the school team<br>-    Establish and facilitate network with external partners<br>-    Building positive relationships with students | "Communicating is an ability without which all other areas are inconceivable." ([68]/20)<br>"Within the institutional and societal settings, teachers must look for cooperation partners within and outside of their own institutions. While these are skills teachers generally need, they are paramount in the complex ESD setting." ([68]/17)<br>"The educator is able to connect with the students and get them to participate in their local and global spheres of influence" ([34]/779). | [1,5,11–13,34,60,66,68] |
| **Knowledge of socio-ecological impact of education**<br>-    Analyse socio-ecological consequences of educational actions<br>-    Plan education with positive socio-ecological impact/mitigate impact through education | "Knows how to develop educational actions that mitigate negative socio-environmental impacts" ([9]/202) | [9,59,63,65] |
| **Sustainable use of resources in the educational setting**<br>-    Act as role model through the sustainable use of resources within the school setting | Understands and integrates the ethical principles of sustainable consumption in his/her actions, considering nature as a good in itself and transmitting the importance of education for a change in the relationship between human beings and the socio-cultural environment" ([63]/766) | [63] |

3.4.1. Beliefs and Values

(a)    Subjective theory and attitude towards SD

Subjective theories guide how teachers think and act in the classroom. In order to create emancipatory ESD, teachers should have an emancipatory idea of SD. Teachers need to "develop their own integrative view of the issues and challenges of SD" [1] (p. 52) and be convinced that SD "is viable" [68] (p. 19) and may only be reached through a societal transformation (Table 5).

(b)    Subjective theory and attitude towards ESD

Several papers described subjective theories and attitudes that are thought to be conducive to an emancipatory ESD. Teachers need to acknowledge the importance of bringing issues of SD into school for tackling societal transformation [66]. UNECE [12] stresses that teachers should understand "why there is a need to transform the way we educate/learn" (p. 14) and "why there is a need to transform the education systems that support learning" (p. 14). This also means opening up schools for external partnerships.

(c)    Epistemological beliefs about knowledge

For an emancipatory ESD epistemological beliefs about knowledge should reflect "the importance of scientific evidence in supporting SD" [12] (p. 14) which nevertheless must be critically challenged. The development of knowledge is embedded in the cultural context and dependent on the values of a society. This dependency should be reflected in the educational context. It should be distinguished from subjective opinions and it must be viewed as dynamic, i.e., preliminary, contradictory and uncertain. It is important to be aware of preconceptions and pre-existing knowledge because they determine worldviews and what and how new knowledge is integrated.

(d)    Self-perception and self-reflection

The reviewed papers suggest that ESD requires a "changing teacher role" [5] (p. 22, author translation). Two focal points are suggested for the teacher role. First, the teacher is expected to be a critically reflective individual; second, the teacher is expected to be "a *critically reflective practitioner*" [12] (p. 14).

**Table 5.** Subcomponents of ESD-specific pedagogical beliefs and values as part of ESD-specific professional action competence.

| Beliefs and Values—Subcategories | Example (Reference/Page) | References |
|---|---|---|
| **Subjective theory of and attitude towards SD** <br><br> - Acknowledge idea of emancipatory SD as important task for society with sense of urgency for change <br> - Recognize that the current cultural development is the root for the ecological crisis <br> - Embrace uncertainty as ethical, social and political attitude <br> - Realize that each person is important and may help | "Acknowledgement of the importance of the regulative idea of SD as a task and a challenge for society as a whole" ([10]/5076) <br> "The educator is able to work from the perspective of uncertainty as an ethical, social and political attitude" ([34]/781) <br> "learning that each person can help improve human development" ([58]/723) | [1,10–12,34,58,66,68] |
| **Subjective theory of and attitude towards ESD** <br><br> - Acknowledge ESD as a resource for tackling transformation towards SD s societal task <br> - Pursue educational goals of an emancipatory ESD: key sustainability competencies <br> - Realize that ESD must build on students' experiences <br> - View ESD as an alternative way of education <br> - Open school for partnerships | "Acknowledgement of the role of education as a resource for the tackling of this societal task" ([10]/5076) <br> "awareness that we currently educate students to reproduce [the] cultural crisis" ([66]/49) <br> "Reflect on the relationship of formal, non-formal and informal learning for sustainable development, and apply this knowledge in their own professional work" ([1]/52) | [1,9–12,34,59,63,66] |
| **Epistemological beliefs about knowledge** <br><br> - Acknowledge that scientific evidence is important to support SD <br> - Knowledge is culture- and value-driven, uncertain, contradictory, preliminary, must be challenged <br> - Knowledge may and must be developed in a joint approach <br> - Pre-existing knowledge determines how we see the world, new knowledge must be integrated into existing knowledge | "The educator . . . knows the importance of scientific evidence in supporting sustainable development" ([12]/14) <br> "becoming aware that knowledge is culture and value driven; tackling the uncertainty, preliminarity and contradictions of such knowledge" ([68]/17). <br> "The educator . . . is inclusive of different disciplines, cultures and perspectives, including indigenous knowledge and worldviews" ([12]/14). <br> "The goal of an educator is to help learners to process new knowledge explicitly and not to simply be exposed to information about the world." ([13]/11) | [1,11,12,34,68] |

| Beliefs and Values—Subcategories | Example (Reference/Page) | References |
|---|---|---|
| **Self-perception/self-reflection**<br>- Self-perception of the teacher role: the teacher as facilitator and participant in the learning process<br>- Personal self-reflection: the teacher as critically reflective individual reflects his/her own lifestyle and decisions, manages and explains own beliefs, values and emotions<br>- Professional self-reflection: the teacher as critically reflective practitioner clarifies his/her own beliefs and values in the teaching context | "The educator is . . . a facilitator and participant in the learning process" ([12]/14)<br>"be able to share the responsibility for the teaching process with learners" ([11]/73)<br>"Teachers have to be aware of the impact of emotions on perception, judgement, decisions and acting in their own lives and the lives of their students and to take account of this in the way they teach ([11]/67)<br>"critically analyses and assesses the consequences his/her [ . . . ] professional actions may have [ . . . ] on promoting sustainable human development" ([59]/5) | [5,9,11,13,34,58,59,61] |

### 3.4.2. Motivational Orientations

(a)  Self-efficacy for teaching ESD

Self-efficacy for teaching ESD was only investigated and discussed in one paper. Malandrakis et al. [65] developed a scale to assess ESD-specific teaching self-efficacy in four subdomains: values and ethics, systems thinking, emotions and action (Table 6).

(b)  Intrinsic motivation and enthusiasm

Three aspects of motivation may be distinguished in the reviewed literature. First, a personal motivation to follow the guiding principle of SD, to challenge assumptions and take action. This may be supported by an optimistic attitude and the conviction that together with others one "can make a contribution toward that end" [68] (p. 19). Second, a professional motivation, i.e., a motivation to develop educational actions for ESD. Since ESD requires effort in terms of cooperation or structural changes, teachers need a certain perseverance, to "keep their enthusiasm for ESD alive" [68] (p. 19) in the school setting and "for lobbying outside the school for ESD" [11] (p. 68). Third, an ability to motivate others, namely, students and colleagues.

### 3.4.3. Self-Regulation

In order to be equipped for the challenge of ESD, teachers need self-regulatory skills. Most papers mentioned self-regulation as an important skill in the context of tackling the problems of SD. However, this aspect of competence is usually described more generally (KSC) and not specifically for the teaching context. We only found one aspect of self-regulation that was particularly mentioned as an important aspect of teacher competence: the ability to express and manage one's emotions and feelings and to "use them constructively to improve situations in the school and community" [11] (p. 67) (Table 6).

**Table 6.** Subcomponents of ESD-specific motivational orientations and self-regulation as part of ESD-specific professional action competence.

| Category | Example (Reference/Page) | References |
|---|---|---|
| **Motivational Orientations** | | |
| **Self-efficacy for teaching ESD**<br>Subdimensions: self-efficacy to . . .<br><br>- Develop values and ethics in students<br>- Foster systems thinking<br>- Develop students' ability to reflect on, express and explain emotions, feelings and empathy<br>- Develop students' ability to initiate actions, reflect on and evaluate them | Confidence to "develop students' VALUES related to sustainable development (e.g., equity, justice, democracy, solidarity, respect to difference)" ([65]/appendix p. 2) | [65] |
| **Intrinsic motivation and enthusiasm**<br><br>- Personal motivation to become sustainable<br>- Professional motivation to initiate education towards sustainable development<br>- Ability to motivate others (students and colleagues) | "The educator . . . is willing to challenge assumptions underlying unsustainable practice; . . . is willing to take action even in situations of uncertainty" ([12]/14)<br>"reasons and motivations to develop projects related to sustainability and social responsibility" ([58]/723).<br>"The educator . . . encourages individuals to become active agents for change" ([34]/780) and "to develop a critical and active society" ([34]/781).<br>"Instead of promoting fears and frustration by doomsday rhetoric, they encourage learners in their commitment" ([68]/19) | [11,12,34,58,68] |
| **Self-regulation** | | |
| Express and manage one's emotions and feelings | "Teachers have to be able to use ways and methods to express and manage their emotions and feelings alone and in groups (e.g., conflict management) and use them constructively for improving situations in the school and community (cultural, ecological, social, economic)." ([11]/67) | [11] |

## 4. Discussion

In the following we will first discuss research questions (1) and (2) based on the results from the systematic review. Subsequently, question (3) will be addressed by mirroring the results of the systematic review with PE-specific literature.

### 4.1. Approaches to Define ESD-Specific Professional Action Competence (RQ 1)

We found five distinct approaches to the definition and structure of ESD-specific professional action competence. They range from a very broad understanding of profession-unspecific KSC to rather narrow profession-specific conceptualizations (EDINSOST, PAC). Most of the scientific debate reflected in this review takes place in Europe. More literature from the USA, Australia and the Asian–Pacific area was detected through a database search but the respective papers did not meet the eligibility criteria. Most papers were excluded from further analysis because they focused on environmental education, without explicitly referring to the multidimensional nature of ESD and often disregarded sociocultural and economic aspects or the concepts of inter- and intragenerational justice (e.g., [71–74]).

The generic model of professional competence (PAC) has predominantly been used in the German-speaking countries (exceptions: USA [66]; Greece [65]) whereas the EDINSOST framework has been only discussed in the Spanish context. Approaches by international teams (UNECE, CSCT and its derivates, e.g., [13,34]) usually are rather complex with extensive (and redundant) lists of subcomponents [11,34] or more concise lists of rather

complex competences [13]. However, empirical evidence suggests that these complex competences are difficult to teach and acquire [75].

The focus of this review was on normative and theoretical concepts of ESD-specific teacher competence. Therefore, it is not surprising that many of the publications reviewed here are normative in nature. Some researchers additionally conducted case studies in which they designed a course about sustainability issues and then evaluated student teachers' learning gains. The quality of these empirical investigations ranges from small studies without control groups and qualitative (e.g., [61]) or mixed-methods designs (e.g., [62]) to more complex quasi-experimental designs (e.g., [69,70]). Empirical evidence regarding the structure of and associations among specific subcomponents of ESD-specific teacher competence are mostly reported in research based on the PAC approach [64,65,69,70]. Looking beyond the basic theoretical and normative work reviewed here, more empirical evidence about the structure and the learning of ESD-specific teacher competence is emerging, especially regarding the RSP-framework [75] and the PAC model [39,76].

*4.2. ESD-Specific Professional Action Competence (RQ2)*

One goal of this review was to compile normative and theoretical approaches to define ESD-specific teacher competence and to summarize subcomponents into one framework based on the generic model of PAC [14]. Some of the existing approaches offer a list of very complex competences (e.g., [13]) that we categorized into several of the subcategories of the refined model of ESD-specific PAC. Others offer extensive lists of specific rather atomic competences (e.g., [11,34]) that we structured within our model based on the theoretical model of PAC. The refined model of ESD-specific PAC that we developed is based on a rather narrow understanding of competence, i.e., it is specific to the teaching profession and the domain of ESD. The subcategories that we inductively developed based on the material are generally in line with the generic model of PAC [14]. The refined ESD-specific model offers a differentiated view on the knowledge bases (CK, PCK, ICK) that are important for teachers in the context of ESD.

Content knowledge (CK) and pedagogical content knowledge (PCK) are conceptualized as domain-specific facets of knowledge in line with Baumert and Kunter [14]. Regarding PCK, we inductively defined ESD-specific teaching principles based on the reviewed literature. These principles fit well into the holism–pluralism–action-orientation paradigm, a teaching approach suggested to support students' acquisition of KSC [27]. For example, holistic teaching would use the knowledge base from multiple disciplines and take into account multiple perspectives (e.g., ecological, socio-cultural, economic, political; time and space). Pluralistic teaching aims to create an appreciative atmosphere respecting diversity where students have the opportunity to develop critical thinking. Action-orientation is reflected in several teaching principles, such as the creation of participative learning environments, the creation of learning tasks from real world problems and the exposure to uncertainty, dilemmas and conflicts of interests. The educational literature refers more generally to the teaching principles we identified with terms like 'enquiry-based learning' [77], 'constructivist teaching' [78], 'reflective teaching' [79] and 'collaborative learning' [80]. These teaching and learning approaches in various nuances are designed to facilitate student-centred, self-directed learning and thus the development of competences in various domains.

Institutional context knowledge (ICK) is interpreted as a domain-specific facet of knowledge (domain: ESD) here, even if the corresponding organizational knowledge in the sense of Baumert and Kunter [14] was conceptualized as domain-unspecific. It is often highlighted that ESD can only succeed when a whole institution approach is implemented in learning institutions [81]. Teachers from different subjects and subject areas usually emphasize different yet complementary aspects of ESD [40]. They need to work together through "cross-curricular teaching" [40] in order to implement ESD in a holistic way in schools. Therefore, teachers need ICK as an ESD-specific facet of knowledge in order to analyse and design not only their own teaching but also their institution.

The relevance of teachers' values and beliefs and how they are used in the classroom is emphasised in the ESD literature. This is because SD issues are not only factually but especially ethically complex [29] and discussions around SD are often value-laden and emotional. What seems to be of particular importance in the context of ESD, are teachers' beliefs with regard to dealing with uncertainty and complexity [28]. Uncertainty and complexity play an important role in modelling teachers' subjective theories and epistemological beliefs in our model (e.g., embrace uncertainty as an ethical, social and political attitude; knowledge is culture- and value-driven, uncertain and contradictory).

ESD deals with different areas of young people's lives and the values and norms of everyday actions from a perspective of SD. Personal views and lifestyle choices are challenged and therefore the lifestyle and personal views of the teacher and her or his role as a model seem to be of particular importance in the ESD context [63]. In some of the reviewed publications, KSCs, i.e., competences a teacher needs to act in favor of SD and to reach out to society, are discussed as key to teaching ESD [12,61,67]. However, in our model, KSCs are not being reflected as specific teacher competences but as overarching or basic competences that radiate into the specific teacher setting via motivational orientations, self-regulation and beliefs.

*4.3. Integration of ESD-Specific Professional Competence in a PE Teacher's Professional Self (RQ3)*

In academic discourse and curricula, physical education (PE) has a dual mission—at least in the German context [82]: education *for* sport aims at performance in specific sport activities and enhancing physical activity using activities that are derived from the sport and exercise context [83,84]; education *through* sport, game and movement aims at personal development and knowledge acquisition [82]. To implement this dual mission, PE teachers need CK about different sport activities or movement areas and related PCK with regard to initiating and accompanying exercise, training, play, problem solving and cognitive processes [37,85]. However, they also need professional knowledge to implement a more holistic pedagogy in PE [86] which provides learners with "new experiences, sometimes with uncertain and surprising outcomes for young people, as well as teachers" (p. 127). Following the theoretical discussion about PE's dual mission, ESD might already be part of PE if the holistic pedagogy includes key sustainability competences (KSCs) as learning goals that are addressed by movement-based tasks and learning activities in the context of sport and physical activity. Empirical evidence suggests that PE teachers are in principle open to the position of education through sport [87]. However, they often tend to follow a conservative position, i.e., education for sport, in their concrete lesson design [87] and they often assume that positive side effects like personal development or social learning are automatically achieved. Therefore, more research and curriculum design in PE teacher education is needed to bring the dual mission of PE from paper (curriculum) and academic discussion to practice.

In order to implement ESD in PE, teachers must acquire and integrate ESD-specific aspects of professional competence in their PE-specific professional selves [3]. In Supplement E we present more details about subcomponents of ESD-specific PAC for the subject PE. The sport- and PE-specific aspects of ESD-related CK, PCK and ICK are based on a range of literature (as displayed in Supplement E). Here, we will only discuss some examples. Regarding CK, PE teachers should have a profound knowledge regarding the challenges and problems of SD in the sport context. This includes knowledge about relevant "inconvenient truths" [88] of globalized sport, e.g., the environmental impact of sport activities and events, climate effects of sport-related mobility, work conditions in the production and supply chain of sport equipment, infrastructure and events [89–92] and also knowledge about potential positive impacts of sport and physical activity on the individual and societal level, e.g., health benefits, sport for peace and development [43,93,94]. Teachers should thus be able to analyse the power and limitations of sport and physical activity as "an important enabler" [95] (para. 37) of the sustainable development goals. A list of SD-related topics that are suitable for fostering KSC through PE but also cross-curricular lessons is

provided by Gieß-Stüber and Thiel [4]. Regarding PCK, PE teachers should be aware of the special possibilities of PE as a subject that relies on physical involvement in contrast and complementary to other (cognitive) subjects. For example, Gieß-Stüber and Thiel [4] demonstrate how role play related to sport games might help to unravel structures of global sports and their effects on the sport activity and its stakeholders. Lohmann et al. [3] show how KSC might be implemented as learning goals in outdoor sport field trips and how socio-cultural, ecological and economic issues of outdoor sport activities might be taught with a collaborative and enquiry-based learning approach using sport tourism as a real-world phenomenon for learning. Grimminger-Seidensticker and Möhwald [47,96] empirically evaluate the Intercultural Movement Education approach [46], a didactical approach for social and intercultural learning based on small games in PE. The examples for the application of ESD-specific principles in the context of PE are based on the literature displayed in Supplement E. Future research is needed in order to systematically draw the lines between PE and ESD on a didactical level.

Global, intercultural and environmental education already seem to have their place in PE. Taken together, these approaches show that ESD goals, content and methods are already being implemented and discussed in PE but without explicitly assigning them to the educational concept of ESD. For this paper we conceptualized ESD as a separate domain for which teachers need to acquire domain-specific PAC. This ESD-specific competence needs to be integrated into the subject (PE)-specific professional competence [3]. One problem with this conceptualization might be a sense of an overcrowded curriculum in which traditional PE and sport contents must be complemented with sustainability topics. However, ESD might also be interpreted as an overarching approach to teaching and learning as it is reflected in German federal curricula [15]. In this sense, ESD-specific professional competence might be conceptualized as a rather general pedagogical competence that serves as a basis for shaping learning environments in PE and in other subjects. Following this argument, PE teachers would not necessarily need to add contents and goals to an existing PE curriculum but rather rethink their teaching from a methodological perspective and reshape the formulation of learning goals in the sense of ESD. Especially in the subject of PE, which is very popular with students, teachers should be able to offer and moderate reflection on SD issues. Through well-guided reflection, students can not only learn to classify what they have experienced in exercises and games but can also be encouraged to become critically reflective individuals in the field of sport, game and movement [97].

### 4.4. Limitations and Future Perspectives

We conducted a database search in English and German that was complemented by a manual search based on a theoretical lens directed at the ESD-specific PAC of teachers. This focus might lead to a geographical bias, since the concept of PAC has been predominantly used in the German literature and the concept of ESD seems to be discussed more deeply in the European context. Future studies might therefore widen the focus to explicitly grasp notions of ESD from other parts of the world and avoid a Eurocentric view.

This review is based on mostly normative contributions to the research of ESD-specific PAC of teachers in various subjects. Through our search we did not identify any PE-specific literature that met the inclusion criteria, i.e., papers about ESD-specific teacher competences. Our work is a normative contribution to the research of ESD-specific PAC and lacks empirical support. However, by placing the results in the didactic literature on PE we show important links to ESD in the discussion and in Supplement E, which contains theoretically derived suggestions by the authors about the integration of ESD-specific competences of PE teachers. We demonstrated that other concepts like intercultural education, environmental education and holistic pedagogical approaches leading to personal development are discussed in the context of PE and might be fruitfully used for the purpose of ESD. A concept for the intentional teaching of ESD hasn't been established so far. More empirical and theoretical research is needed in order to elucidate the role of PE and PE teacher education

for the implementation of ESD in the education system and how respective ESD-specific competences may be fostered through PE and PE teacher education.

## 5. Conclusions

This systematic review offers a comprehensive literature review of approaches to describe, define and research ESD-specific teacher competence. Based on the theoretically and empirically sound framework of professional action competence (PAC) [14,32,35], we summarized the subcomponents of existing frameworks and suggested a refined model of ESD-specific PAC for teachers based on the current body of literature. The discussion demonstrates how this model might be applied to subject-specific discourses using the example of physical education (PE). We hope that the suggested model of ESD-specific PAC is helpful for future theory-driven intervention and implementation studies in (PE) teacher education and may serve for curriculum development. With this work we provide a theoretical and normative contribution to the research of ESD-specific teacher competence so that ESD may become even more viable through profound teacher education in the future.

**Supplementary Materials:** The following are available online at https://www.mdpi.com/article/10.3390/su132313343/s1, Supplement A: Search strategy; Supplement B: Critical appraisal checklist; Supplement C: Quantitative results: Number of codings per category in each paper; Supplement D: ESD-specific teaching principles; Supplement E: PE teachers ESD-specific professional action competence.

**Author Contributions:** Conceptualization, J.L., J.B., U.O., P.G.-S. and H.P.B.-B.; methodology, J.L. and J.B.; validation, J.L. and J.B.; formal analysis, J.L., J.B.; investigation, J.L. and J.B.; data curation, J.L.; writing—original draft preparation, J.L. and J.B.; writing—review and editing, J.L., J.B., U.O., P.G.-S. and H.P.B.-B.; visualization, J.L.; supervision, U.O., P.G.-S. and H.P.B.-B.; project administration, J.L.; funding acquisition, J.L. and H.P.B.-B. All authors have read and agreed to the published version of the manuscript.

**Funding:** This research was funded by the Federal Ministry of Education and Research grant number 01JA1809. The article processing charge was funded by the Baden-Wuerttemberg Ministry of Science, Research and Art and the University of Freiburg in the funding program Open Access Publishing.

**Institutional Review Board Statement:** Not applicable.

**Informed Consent Statement:** Not applicable.

**Acknowledgments:** We want to thank Hanna Kolb (H.K.) and Marianne Wachsmann (M.W.) for their supporting work during the data analysis (screening of titles, abstracts and full texts).

**Conflicts of Interest:** The authors declare no conflict of interest.

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
