# Peer review of "Teachers’ Professional Action Competence in Education for Sustainable Development: A Systematic Review from the Perspective of Physical Education"

_sustainability, doi:10.3390/su132313343_

Round 1

Reviewer 1 Report

The article defines and addresses a clear and interesting challenge. This is a very well-structured, well-written paper in that procedures and rationales for procedures were clearly explained. The foci of the article and the need for the study were similarly clearly set out. The Tables summarise findings and add clarity. Careful analysis of the literature resulted in useful categories and findings.

Given the ESD context and that not all (for example, PE educators) reading the article will be familiar with the specificities of ESD, I recommend including a clear statement related to environmental (or not in this case) and equity agendas of ESD in the Introduction at the outset.

Minor style and language checks more for aesthetics and formality/non-colloquialism than anything else, for example, " overview over" in Line 59, or "meant to be" in line 68. Address the issue in line 320. 

If inclusion criteria were from the year 2000, why was The Journal of teacher education for sustainable development searched only for the years 2018 - 2020? 

There are many acronyms - maybe helpful for the reader to re-include the full term at strategic points.

Author Response

Response Letter

Manuscript ID: sustainability-1473824

Title: Teachers’ professional action competence for education for sustainable development: A systematic review from the perspective of physical education

Dear reviewers,

first of all, we would like to thank you for taking the time to assess our manuscript and for providing us with such a thorough review. In order to address all of your comments, we have responded to them in detail in the following text.

Overall, we tried to follow the recommendations by adding further information to the manuscript and reworking it. In a few specific cases, we have justified why we want to stick to the original text passage.

Kind regards,

The authors

***

Reviewer 1

The article defines and addresses a clear and interesting challenge. This is a very well-structured, well-written paper in that procedures and rationales for procedures were clearly explained. The foci of the article and the need for the study were similarly clearly set out. The Tables summarise findings and add clarity. Careful analysis of the literature resulted in useful categories and findings.

Thank you for your positive evaluation of our manuscript and your comments. You will find a point-by-point reply to your comments below.

# 1

Given the ESD context and that not all (for example, PE educators) reading the article will be familiar with the specificities of ESD, I recommend including a clear statement related to environmental (or not in this case) and equity agendas of ESD in the Introduction at the outset.

Reply: Thank you for this advice. We included one sentence regarding environmental and global citizenship education in section 1.3.

# 2

Minor style and language checks more for aesthetics and formality/non-colloquialism than anything else, for example, " overview over" in Line 59, or "meant to be" in line 68. Address the issue in line 320. 

Reply: We did a minor style and language check with an English native speaker with the revised manuscript. We hope that this improved the aesthetics of our manuscript. [see markup version for any changes].

# 3

If inclusion criteria were from the year 2000, why was The Journal of teacher education for sustainable development searched only for the years 2018 - 2020? 

Reply: The reason for this manual search was to doublecheck the database search. We wanted to be sure that the database search revealed relevant papers. Therefore, we wanted to see, if a manual search would reveal any ADDITIONAL papers – which it did not. For pragmatic reasons, we only manually searched the most recent years. We added the respective information in Line 321.

# 4

There are many acronyms - maybe helpful for the reader to re-include the full term at strategic points.

Reply: Thank you for this advice. We reviewed the titles of chapters and first mentions of acronyms within chapters and included the full terms at strategic points (e.g., in the data analysis section)

Reviewer 2

I thank the authors for their impressive research. The problem covered is indeed important, and it needs an in-depth approach. Review of existing studies can help with this. I agree with authors that often it is difficult to match ESD with specific subjects. I think the authors took a very smart approach by focusing on ‘seemingly’ unmatching discipline (PE) and ESD. I agree with authors and their citing source that physical activities ‘may be the starting point for differentiated experiences with and reflections about individual sustainable lifestyle choices relating to physical activity.’ Well done.

Thank you for your positive evaluation of our manuscript and your valuable comments. You will find a point-by-point reply to your comments below.

I have a few comments and one critical point (last one) to help authors improve this already excellent research, as follows:

# 1

In section 1.1. Researchers have effectively outlined the educational policy framework for ESD in Germany – would it be possible to add a few paragraphs to show differences and similarities with other EU or non-EU countries?

Reply: Thank your for this comment! The international dimension is indeed very interesting. To integrate it, we have included some paragraphs on international documents in section 1.1. These documents show important focal points for the implementation of ESD on the european and global level. A differentiated comparison of educational policy frameworks for ESD in EU and non-EU countries in terms of similarities and differences would be very interesting. But this would require in-depth analysis and further systematic review of documents and literature. This would go beyond the scope of this article.

 # 2

The section 1.4. is titled ‘Existing frameworks for teachers’ ESD-specific professional competence.’ However, reading further I feel that what authors write about are ‘approaches’ (i.e. belief-based, professional knowledge-based approaches) not necessarily ‘framework.’ In the context of ESD one can get easily confused because framework implies some empirical formality and structure, whereas approaches are more flexible and theoretically malleable. By the way, the last paragraph of this very section also mentions ‘various approaches to ESD-specific teacher competence’’. The authors need not only consistency in language use but also approach words carefully as this can backfire in the end.

Reply: Thank you for this valuable comment. We revised the text, using “approaches to define competence” when we write about more general approaches, and “framework” if we refer to specific (empirically tested/theoretically derived) competence frameworks.

 # 3

For transparency and visualization purposes, the authors should provide more some detailed rationale (reasons) for excluding 565 records and 113 full text articles. This can be solved easily by adding bullet points inside corresponding boxes in the flow chart.

Reply: Thank you for this suggestion. We added some bullet points for explaining the exclusion criteria in the flow chart, the most important were:

  • Papers did not make a theoretical or normative contribution to research in ESD-specific teacher competence
  • Papers focused on EE or GCE (rather unidimensional concepts) instead of holistic concept of ESD
  • Competences were formulated at the pupils or teacher educators’ level or for students in general, and did not specifically address (prospective) teachers

# 4 

I have some concerns about the authors approach to Appraising the quality of included studies (Section 2.2.). To me, categorizing by ‘low, medium and high’ is somewhat subjective (Table 1). This approach would perhaps work in literature review for an empirical paper, but this is a systematic review. The most important aspect that differentiates systematic reviews from a conventional review is the extend to which bias was eliminated in the selection and appraisal stages, in the former’s case. I strongly recommend the authors to double check their appraisal procedures using more formal instruments (checklists) such as Critical Appraisal Skills Program (CASP). Please check the website (https://casp-uk.net/casp-tools-checklists/) to access from a number of instruments that fits your paper. Since you are using qualitative content analytic design you might find this instrument more useful - https://casp-uk.b-cdn.net/wp-content/uploads/2018/03/CASP-Qualitative-Checklist-2018_fillable_form.pdf

Or this one (for Systematic Reviews in general) https://casp-uk.b-cdn.net/wp-content/uploads/2018/03/CASP-Systematic-Review-Checklist-2018_fillable-form.pdf

Reply: Thank you for this valuable comment. We have reviewed the proposed lists. In our review, we are interested in the theoretical and normative concepts of ESD-specific teacher competences. Therefore, we could not use any of the CASP-checklists for our quality appraisal The CASP-lists are for the quality appraisal of EMPIRICAL qualitative and quantitative studies or systematic reviews. 

However, we revised our quality appraisal according to your suggestions referring to the JBI critical appraisal checklist for text and opinion papers (JOANNA BRIGGES INSTITUTE, 202,0 HTTPS://JBI.GLOBAL/CRITICAL-APPRAISAL-TOOLS). As this checklist does also not fit our purpose entirely, we adapted the criteria to our goals and content – as displayed in the new supplement B.

The rating was conducted by JL and JB independently. Inconsistencies between the two raters were discussed and resolved by consensus. The revised results of quality appraisal are displayed in table one. We used the quality appraisal for included studies to inform the reader about the quality of the reviewed papers. Therefore, we would like to keep the structure with three levels of elaboration as we suggested it in the first draft.

# 5 

What authors could do is rename their Table 1 column titled ‘Quality’ with ‘Quality appraisal with CASP’ and instead of writing ‘low, medium, high’ they could type ‘passed’ if the selected articles indeed passes all the quality checklists (if at least two researchers could do that, it would make the appraisal much more transparent and objective).     

Reply: We renamed the column title. The new column title is “quality appraisal”. As a note to the table we added information about the quality appraisal (see also Supplement B).    

# 6 

Mentioning CASP in the main text could also increase the acceptance of their authors results.

Reply: s.o.: Please see our comments above. We did not use the CASP lists because they are not appropriate for our purpose. We refer to the critical appraisal list by JBI and also mentioned this in the text (see section 2.2 and note for Table 1)

I am satisfied with the rest of the manuscript.

Thank you very much for your work with our manuscript and your general positive evaluation.

Reviewer 2 Report

I thank the authors for their impressive research. The problem covered is indeed important, and it needs an in-depth approach. Review of existing studies can help with this. I agree with authors that often it is difficult to match ESD with specific subjects. I think the authors took a very smart approach by focusing on ‘seemingly’ unmatching discipline (PE) and ESD. I agree with authors and their citing source that physical activities ‘may be the starting point for differentiated experiences with and reflections about individual sustainable lifestyle choices relating to physical activity.’ Well done.

I have a few comments and one critical point (last one) to help authors improve this already excellent research, as follows:

  1. In section 1.1. Researchers have effectively outlined the educational policy framework for ESD in Germany – would it be possible to add a few paragraphs to show differences and similarities with other EU or non-EU countries?

  1. The section 1.4. is titled ‘Existing frameworks for teachers’ ESD-specific professional competence.’ However, reading further I feel that what authors write about are ‘approaches’ (i.e. belief-based, professional knowledge-based approaches) not necessarily ‘framework.’ In the context of ESD one can get easily confused because framework implies some empirical formality and structure, whereas approaches are more flexible and theoretically malleable. By the way, the last paragraph of this very section also mentions ‘various approaches to ESD-specific teacher competence’’. The authors need not only consistency in language use but also approach words carefully as this can backfire in the end.

  1. For transparency and visualization purposes, the authors should provide more some detailed rationale (reasons) for excluding 565 records and 113 full text articles. This can be solved easily by adding bullet points inside corresponding boxes in the flow chart.

  1. I have some concerns about the authors approach to Appraising the quality of included studies (Section 2.2.). To me, categorizing by ‘low, medium and high’ is somewhat subjective (Table 1). This approach would perhaps work in literature review for an empirical paper, but this is a systematic review. The most important aspect that differentiates systematic reviews from a conventional review is the extend to which bias was eliminated in the selection and appraisal stages, in the former’s case. I strongly recommend the authors to double check their appraisal procedures using more formal instruments (checklists) such as Critical Appraisal Skills Program (CASP). Please check the website (https://casp-uk.net/casp-tools-checklists/) to access from a number of instruments that fits your paper. Since you are using qualitative content analytic design you might find this instrument more useful - https://casp-uk.b-cdn.net/wp-content/uploads/2018/03/CASP-Qualitative-Checklist-2018_fillable_form.pdf

Or this one (for Systematic Reviews in general) https://casp-uk.b-cdn.net/wp-content/uploads/2018/03/CASP-Systematic-Review-Checklist-2018_fillable-form.pdf

What authors could do is rename their Table 1 column titled ‘Quality’ with ‘Quality appraisal with CASP’ and instead of writing ‘low, medium, high’ they could type ‘passed’ if the selected articles indeed passes all the quality checklists (if at least two researchers could do that, it would make the appraisal much more transparent and objective).         

Mentioning CASP in the main text could also increase the acceptance of their authors results.

I am satisfied with the rest of the manuscript.

Author Response

(The authors gave the same response as above.)
